# Night-time NO emissions strongly suppress chlorine and nitrate radical formation during the winter in Delhi

Sophie L. Haslett[1,2], David M. Bell[3], Varun Kumar[3†], Jay G. Slowik[3], Dongyu S. Wang[3], Suneeti Mishra[4], Neeraj Rastogi[5], Atinderpal Singh[5‡], Dilip Ganguly[6], Joel Thornton[7], Feixue Zheng[8], Yuanyuan Li[9], Wei Nie[9], Yongchun Liu[8], Wei Ma[8], Chao Yan[10], Markku Kulmala[8,9,10], Kaspar R. Daellenbach[10], David Hadden[1,2], Urs Baltensperger[3], Andre S. H. Prevot[3], Sachchida N. Tripathi[4] and Claudia Mohr[1,2§]

[1]Department of Environmental Science, Stockholm University; Stockholm, Sweden.
[2]Bolin Centre for Climate Research; Stockholm, Sweden.
[3]Laboratory of Atmospheric Chemistry, Paul Scherrer Institute; 5232 Villigen PSI, Switzerland.
[4]Department of Civil Engineering, Indian Institute of Technology Kanpur; Kanpur, India.
[5]Geosciences Division, Physical Research Laboratory; Ahmedabad, India.
[6]Centre for Atmospheric Sciences, Indian Institute of Technology Delhi; New Delhi, India.
[7]Department of Atmospheric Sciences, University of Washington; Seattle, USA.
[8]Aerosol and Haze Laboratory, Beijing Advanced Innovation Centre for Soft Matter Science and Engineering, Beijing University of Chemical Technology; Beijing, 100029, China.
[9]Joint International Research Laboratory of Atmospheric and Earth System Sciences, School of Atmospheric Sciences, Nanjing University; 210023 Nanjing, China.
[10]Institute for Atmospheric and Earth System Research/Physics, Faculty of Science, University of Helsinki; Helsinki, Finland.

*Correspondence to*: Claudia Mohr (claudia.mohr@psi.ch), Sachchida Tripathi (snt@iitk.ac.in), Sophie Haslett (sophie.haslett@aces.su.se)

[†]Now at: Institut National de l'Environnement Industriel et des Risques (INERIS), Parc Technologique ALATA, 60550 Vemeuil-en-Halatte, France.

[‡]Now at: Department of Environmental Studies, University of Delhi; Delhi 110007, India.

[§]Now at: Laboratory of Atmospheric Chemistry, Paul Scherrer Institute; 5232 Villigen PSI, Switzerland

**Abstract.** Atmospheric pollution in urban regions is highly influenced by oxidants due to their important role in the formation of secondary organic aerosol (SOA) and smog. These include the nitrate radical ($NO_3$), which is typically considered a night-time oxidant, and the chlorine radical (Cl), an extremely potent oxidant that can be released in the morning in chloride-rich environments as a result of nocturnal build-up of nitryl chloride ($ClNO_2$). Chloride makes up a higher percentage of particulate matter in Delhi than has been observed anywhere else in the world, which results in Cl having an unusually strong influence in this city. Here, we present observations and model results revealing that atmospheric chemistry in Delhi exhibits an unusual diel cycle, controlled by high concentrations of NO during the night. As a result of this, the formation of both $NO_3$ and dinitrogen pentoxide ($N_2O_5$), a precursor of $ClNO_2$ and thus Cl, are suppressed at night and increase to unusually high levels during the day. Our results indicate that a substantial reduction in night-time NO has the potential to increase both nocturnal oxidation via $NO_3$ and the production of Cl during the day.

**1 Introduction**

Delhi is one of the world's most polluted cities, and experiences its most severe pollution episodes during winter due to stagnant meteorology and a compressed boundary layer (Guttikunda and Gurjar, 2012). Industry, agricultural burning, brick kilns and traffic contribute substantially towards urban haze, in addition to widespread small-scale sources such as cooking emissions, and waste and fuel burning (Rai et al., 2020; Guttikunda and Calori, 2013; Lalchandani et al., 2021; Pant et al., 2015). Many of these are known to emit large quantities of chlorine (Rai et al., 2020; Gunthe et al., 2021; Zhang et al.,

45  2022).

Consequently, particulate chloride concentrations in Delhi are higher than anywhere else in the world where measurements have been made. Chloride makes up around 10% of sub-micron particulate matter by mass (Pant et al., 2015; Gunthe et al., 2021; Gani et al., 2019; Tobler et al., 2020), and from our observations as much as 38% in some cases, compared with 4%

during the winter in Beijing (Zhang et al., 2020). The ramifications of this are only now beginning to be understood. For example, new research has shown that chloride significantly impacts haze and fog formation in Delhi as a result of enhanced water uptake (Gunthe et al., 2021). The severe consequences for human health are made all the more critical by the city's extremely high population density (11,000 people km$^{-2}$ in 2011 (Joshi, 2011)).

High levels of particulate chloride can result in production of the chlorine radical (Cl), a highly reactive oxidant. Many volatile organic compounds (VOCs) are oxidised by Cl at rates far exceeding that of the hydroxyl radical (OH) (Spicer et al., 1998; Osthoff et al., 2008), the main atmospheric oxidant, meaning that even low Cl concentrations substantially increase atmospheric reactivity. This can in turn increase secondary organic aerosol, SOA (Dhulipala et al., 2019; Wang and Hildebrandt Ruiz, 2018) and therefore urban haze formation.


In the urban environment, Cl is extracted from particle-phase chloride when both nitrogen dioxide ($NO_2$) and ozone ($O_3$) are present. Reactions between $NO_2$ and $O_3$ produce the nitrate radical ($NO_3$), another prominent atmospheric oxidant, which forms an equilibrium with dinitrogen pentoxide ($N_2O_5$). Heterogeneous reactions between gaseous $N_2O_5$ and particulate chloride produce nitryl chloride ($ClNO_2$). Due to both the instability of $NO_3$ in daylight and the fast reaction between $NO_3$ and

NO (Table S1), $N_2O_5$ and $NO_3$ are commonly depleted during the day and increase at night (Wang et al., 2017). This pattern promotes the accumulation of night-time $ClNO_2$. After sunrise, $ClNO_2$ is photolysed and Cl liberated.

Here, we present evidence that this diel cycle is inverted in Delhi, with $NO_3$ and $N_2O_5$ being present primarily during the day. Observations of $N_2O_5$ and $ClNO_2$ were carried out using an iodide chemical ionisation mass spectrometer fitted with a filter

inlet for gases and aerosols (FIGAERO-CIMS) in January and February 2019, as part of a larger-scale effort to characterise physical and chemical properties of Delhi's urban haze (Lalchandani et al., 2021; Rai et al., 2020; Singh et al., 2021a; Kumar et al., 2022). During the campaign, we observed particulate chloride loadings in excess of 100 µg m$^{-3}$ on several occasions. Almost no $N_2O_5$ was observed at night, and daytime concentrations were in large excess compared with the few other places where sustained daytime $N_2O_5$ has been observed (Houston, Texas (Geyer et al., 2003) and the Gulf of Maine (Osthoff et al.,

2006)). Consequently, the concentrations of $ClNO_2$ formed were much lower than might have been expected considering the high chloride loadings.

## 2 Methods

### 2.1 Field campaign and sampling site

The field study was carried out at the Centre for Atmospheric Sciences at the Indian Institute of Technology in New Delhi

(IITD; 28.54°N, 77.19°E) during January and February 2019. Measurements were taken from 11 January – 5 February. Instruments were installed in a temperature-controlled fourth-floor laboratory.

The site is surrounded by a mixture of residential, commercial and educational buildings. It is situated in the IITD campus, which also contains a number of green spaces. Delhi's outer ring-road is located around 80 m to the north of the site, and a

larger arterial road can be found around 1 km ESE of the site. The site is a representative background urban site. Measurements took place during the winter season, with temperatures ranging from around 10 °C during the coolest night to 30 °C on the warmest day. Relative humidity ranged from 30% to 100%. More detail on the campaign site itself and the aerosol and gas-phase instrumentation can be found in other publications (Lalchandani et al., 2021; Rai et al., 2020; Singh et al., 2021b; Wang et al., 2020; Kumar et al., 2022).

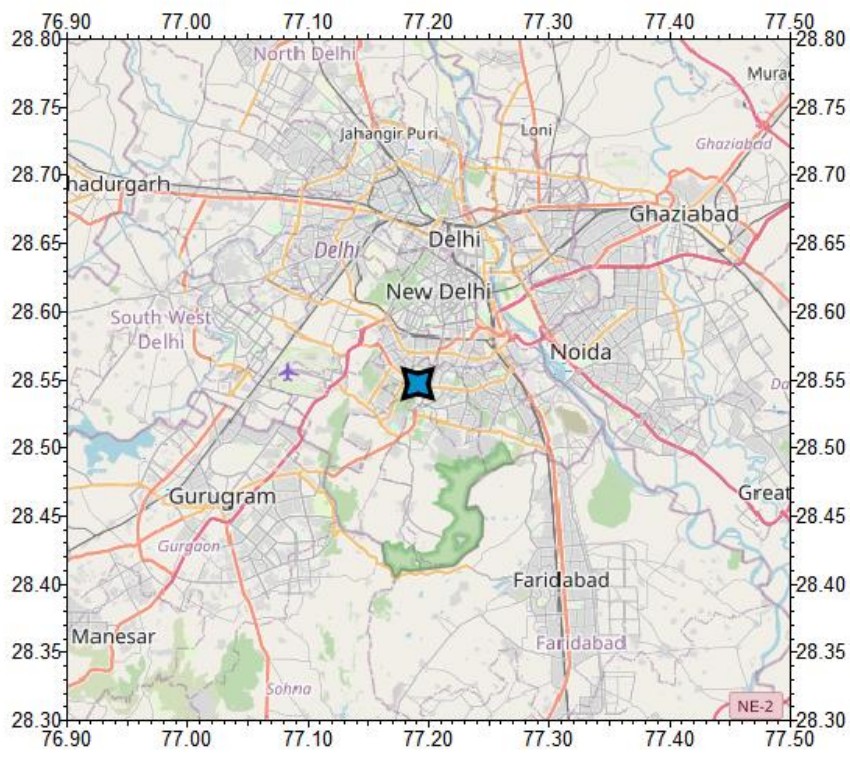

**Figure 1: Map of the field site location at the Indian Institute of Technology in Delhi. (Map data © OpenStreetMap 2023. Distributed under the Open Data Commons Open Database License (OdbL) v1.0.)**

## 2.2 The FIGAERO-CIMS

A high-resolution iodide-adduct chemical ionisation mass spectrometer with a filter inlet for particles and gases (FIGAERO-CIMS) (Lopez-Hilfiker et al., 2014) (Aerodyne Research, Billerica) was used to measure $N_2O_5$, $ClNO_2$ and oxygenated organic compounds. This method uses the negative iodide ion ($I^-$) as a reagent, which is produced by passing a dry nitrogen flow (1.5 lpm) over a methyl-iodide permeation tube, followed by an X-ray ioniser. This ionised ion flow interacts with incoming sample air in the ion molecular region (IMR), which is kept at 250 mbar, creating charged products that are identified as clusters containing $I^-$ in the time-of-flight mass spectrometer. The FIGAERO inlet allows the gas phase to be sampled while simultaneously collecting particles on a Teflon filter (PALL Zefluor, 2 µm pore size, 25 mm diameter). This filter was exposed to a sample air flow of 2 lpm for approximately 3 minutes, before being moved into the desorption position at the instrument inlet. In this position, it was heated gradually from room temperature to 200 °C, using a dry nitrogen flow at 2 lpm, before being held at 200 °C for 20 minutes and then cooled back down to room temperature over the course of 15 minutes. During the heating process, compounds that vaporise at temperatures below 200 °C desorb from the filter, with the quantity of each compound growing as the temperature increases and then peaking at the temperature $T_{max,i}$, which differs for each compound

*i*. The temperature at which the maximum amount of a compound desorbs from the filter is related to the compound's volatility (Lopez-Hilfiker et al., 2014; Thornton et al., 2020). The total signal for each compound from each heating event was found by integrating the signal from the beginning of the heating period to the beginning of the cooling period.

Between particle-phase desorption periods, the gas phase was sampled directly by the FIGAERO-CIMS for approximately 20 minutes. For this study, these 20-minute periods have been averaged, providing averaged datapoints at a frequency of one every 60-90 minutes while the instrument was running. Air samples were drawn at 3.5 lpm through the inlet, approximately 2 lpm of which entered the IMR, with the rest being included only to reduce residence time, and so pumped away. The $N_2O_5$ and $ClNO_2$ ions were observed clustered with $I^-$ at 235 amu (*m/z* 235) and *m/z* 208, respectively (Slusher et al., 2004; Kercher
et al., 2009).

Background measurements were carried out 26 times for the particle phase and 6 times for the gas phase throughout the campaign. For the particle phase, a clean, pre-heated Teflon filter (PALL Zefluor, 2 µm pore size, 25 mm diameter) was inserted into the nitrogen flow and the standard heating procedure described above was carried out. The results from these
gave an indication of the instrument background, and the integrated signal for each compound was subtracted from the integrated signal for each particle-phase measurement. In the gas phase, zero air gas was passed directly into the inlet and the background was measured.

Particle-phase sampling for the FIGAERO-CIMS was conducted via a ½-inch outer diameter (0.35 inch inner diameter)
stainless steel tube of 1.5 m length (2 lpm, 2.8 s residence time) and gas-phase sampling was via a ¼-inch outer diameter (0.2 inch inner diameter) polyfluoroalkoxy- (PFA-) Teflon tube of around 2 m length (2 lpm, 1.2 s residence time).

Direct calibrations for $N_2O_5$ and $ClNO_2$ were not carried out in the field. The method used to quantify the FIGAERO-CIMS signal for these compounds is detailed in the Supplementary Text S1 and Figs. S1 and S2.

**2.3 Other instrumentation**

An aerosol mass spectrometer (AMS; Aerodyne Research, Billerica) was used to quantify aerosol chloride, nitrate, ammonium, sulfate and organic concentrations. The instrument setup for this campaign has been described in detail elsewhere (Kumar et al., 2022). In brief, ambient air was sampled through an aerodynamic lens that limits sampling to particulate matter with a diameter smaller than 1 µm ($PM_1$). Aerosol particles within the beam are flash vaporised at 600 °C, and the resulting vapours
ionised using electron impact ionisation. Ionised fragments pass into the time-of-flight mass spectrometer chamber and are detected using a multi-channel plate (MCP) detector. This method can provide a quantitative measure of each species' concentration in µg m$^{-3}$. Ionisation efficiency calibrations were carried out before and after the campaign using 300-nm

ammonium nitrate particles. More information on this instrument's operation during this campaign has been reported by (Kumar et al., 2022).


NO$_x$ measurements were carried out using a Serinus 40 Oxides of Nitrogen analyser (Ecotech). This instrument was calibrated at multiple concentration levels using the calibrator and standard gas cylinder before and after the campaign. The limit of detection was around 0.4 ppbv. A scanning mobility particle sizer (SMPS, GRIMM) provided size-binned measurements of the dry fine aerosol number concentration between 19 nm and 1 µm. A proton-transfer-reaction time-of-flight mass

spectrometer (PTR, Ionicon Analytical G.m.b.H, Innsbruck, Austria) was used to measure the larger VOCs, and smaller VOCs were observed using an AirmoVOC C2-C6 analyser, model A22022 (Chromatotec ®, France).

Sampling for the NO$_x$ analyser, O$_3$ analyser (Model 202, 2B Technologies) and SMPS was conducted via a 6 mm inner diameter stainless steel inlet of 3 m length, with an ambient particulate matter (APM$_{2.5}$) cyclone (BGI, Mesa Labs. Inc.)

installed to remove larger particles (Kumar et al., 2022). The inlets extended around 1.5 m horizontally from the building's NNW side, around 12 m above the ground. Global radiation was measured on the roof of the building and other meteorological parameters including temperature, relative humidity, wind speed and wind direction were supplied by the Indian Meteorological Department (IMD). The measurement location for these was Indira Gandhi International Airport, which is situated approximately 8 km to the west of our field site.

**2.4 0-dimensional chemical box model**

A simple 0-dimensional (0-D) chemical box model was constructed in order to investigate the relative influence of N$_2$O$_5$ sources and sinks. The model was used to calculate concentrations of N$_2$O$_5$ and NO$_3$ based on values in the previous step, with a time step of 0.04 s and a "spin up" period of 24 hours. The reactions included in the model are outlined in Table S1 in the Supplement (Brown et al., 2003; Yan et al., 2019).


Here, $k_{VOC,i}$ corresponds to the reaction rate constant of each individual volatile organic compound (VOC) in s$^{-1}$, and [VOC]$_i$ to the concentration of the VOC, as measured by the PTR. The heterogeneous reaction rates for N$_2$O$_5$ and NO$_3$ at particle surfaces (R6 and R7 in the supplement) rely on $c_{N2O5}$ and $c_{NO3}$, the average molecular speed of an N$_2$O$_5$ or NO$_3$ molecule ($\bar{c}_X$), the uptake coefficients for N$_2$O$_5$ or NO$_3$, ($\gamma_{N2O5}$ and $\gamma_{NO3}$, respectively) and the available aerosol surface area (S$_A$). These last

two will be outlined in more detail in the supplement. The molecular speeds $\bar{c}_X$ were calculated for each species $X$ using Eq. 1 (Morgan et al., 2015), where $k$ is Boltzmann's constant (1.38e-23 m$^2$ kg s$^{-2}$ K$^{-1}$), $T$ represents the temperature in K and $M_W$ is the molecular weight.

$$\bar{c}_X = \sqrt{\frac{8kT}{\pi M_W}} \qquad\qquad (1)$$

Box model results were compared with results from a simple steady-state approach. Using the steady state approximation, it was possible to calculate $NO_3$ concentrations using Eq. 2 and $N_2O_5$ concentrations using Eq. 3 (Osthoff et al., 2006; Brown et al., 2005).

$$[NO_3]_{calc} = \frac{k_1[NO_2][O_3]}{k_3[NO]+j_4+k_5+k_6K_{eq}[NO_2]+k_7} \tag{2}$$

$$[N_2O_5]_{calc} = K_{eq}[NO_2][NO_3]_{calc} \tag{3}$$

Here, square brackets denote the concentration of the respective compound in molecules cm$^{-3}$. Calculated concentrations of both species were found to agree closely with results from the box model, indicating that the steady-state approximation is reasonable in this case. Therefore, the simpler steady-state approach was used for the results presented here.

Time series data of NO, $NO_3$, $O_3$, VOCs, global radiation, temperature and wet particle surface area are used as direct inputs for reactions R1-R7 (see Supplement). The chemical compositions of VOCs included in the model were $C_2H_6$, $C_2H_4$, $C_3H_8$, $C_4H_{10}$, $C_6H_{14}$, $C_6H_6$, $C_7H_8$, $C_8H_{10}$, $C_9H_{12}$, $C_5H_8$ and $C_{10}H_{16}$, which are all common urban VOCs that were observed in Delhi by the PTR. Measurements were not available for some of the more abundant VOCs in Delhi such as methanol and acetone (Tripathi et al., 2022), and as such, the magnitude of the VOC sink is likely an underestimate. Nevertheless, this sink was minimal compared with the $NO_3$ loss via interaction with NO and photolysis. This underestimate is therefore unlikely to result in substantial impact on the estimated $N_2O_5$ concentration. This model does not account for effects due to dilution, mixing or other atmospheric dynamics. However, much of this will already be accounted for due to changes in the input parameters (NO, $NO_3$, $O_3$ and VOCs). More details about the calculation of various input parameters can be found in the Supplement. The planetary boundary layer height (PBLH) displayed in Fig. 2 and the friction velocity ($U^*$) were obtained from the Real-time Environmental Applications and Display sYstem (READY; Rolph et al., 2017) website, and was available at 3 h resolution.

The campaign averaged diel cycles were used for each input parameter. These diel averages for each compound were constructed by binning all measurements according to the time at which they were taken, and calculating the mean for each hour of the day across the whole campaign.

## 2.5 The Framework for 0-D Atmospheric Modelling (F0AM)

The F0AM model (Wolfe et al., 2016) is a 0-dimensional box model that can be used to simulate atmospheric chemistry systems. Here, we used the Master Chemical Mechanism (MCM) 3.3.1 chemistry, which was extended to include reactions for chlorine chemistry (Riedel et al., 2014). The latter included reactions of Cl with a series of VOCs, and further reactions of their oxidation products.

Observed values of $NO_x$, CO, HONO, $HNO_3$, $ClNO_2$, $ClONO_2$, HOCl, $Cl_2$ and a series of VOCs (Fig. S4, Table S2) were used as model inputs. A 24-hour diel cycle was modelled from 00:00, with a 48-hour "spin up" period. The model produced one datapoint per hour. We used the 'MCM' radiation model, which represents "typical" tropospheric conditions, but does not reflect variability in the ozone column, surface albedo, aerosol optical depth and clouds (Wolfe et al., 2016). The modelled diel cycle of $O_3$ was used to scale the model's photolysis parameter by comparing the calculated $O_3$ with observed values, which resulted in a photolysis scaling factor of 0.25.

The model was used to estimate the concentrations of the oxidants OH, $NO_3$, and $O_3$, and to explore the relative importance of different oxidation compounds for oxidation of different VOCs, with varying contributions from Cl.

## 3. Results and discussion

### 3.1 Campaign overview

Figure 2 shows a time series of meteorological conditions and pollutant concentrations including $NO_x$, $O_3$, $ClNO_2$, $N_2O_5$ and particulate organics, nitrate and chloride during the field campaign in January and February 2019. During this period, the average daily maximum temperature was 22.7 °C, with a range from 19.0 to 31.1 °C, and the average night-time minimum temperature was 12.0 °C, ranging from 9.3 to 16.2 °C. Temperatures were reasonably stable throughout the campaign, with a slight peak around the 20[th] January. Relative humidities were higher at night, with an average low of 46.8% during the day and high of 89.1% at night. For these calculations, an average campaign sunrise time of 07:12 and sunset of 17:53 were used to split the data into daytime and night-time values.

Ozone concentrations reached an average 43 ppbv during the day, but were very low at night, usually just 2 or 3 ppbv. Concentrations of NO, in contrast, were highest during the night, with a mean of 84 ppbv in comparison with 22 ppbv during the day. This is an unusual feature and is likely a consequence of more heavy-duty vehicles using the roads at night due to daytime restrictions (Tobler et al., 2020), combined with the low night-time boundary layer height (Raj et al., 2021) and low nocturnal $O_3$ levels limiting the conversion of NO to $NO_2$. Less diurnal difference was shown by $NO_2$, which averaged around 36 ppbv throughout the campaign, with small peaks in the morning and the evening. Particulate organic concentrations were very high during night-time periods, regularly exceeding 200 µg m$^{-3}$, and reducing to an average of 61 µg m$^{-2}$ during the day, primarily as a result of dilution caused by the change in boundary layer height. Particulate chloride was enhanced in the early mornings, sometimes exceeding concentrations of 100 µg m$^{-3}$. These values are consistent with previous studies carried out in Delhi, which have shown the city to have among the highest levels of particulate chloride measured anywhere in the world (Gani et al., 2019; Gunthe et al., 2021).

Similarly high concentrations of NO and low concentrations of $O_3$ at night have been observed in previous wintertime studies in Delhi. For example, Nelson et al., (2021, 2023) observed a very similar diel NO and $O_3$ pattern to that displayed here in October-November 2018. In a year-long study, Sharma et al., (2021) demonstrated that these high night-time NOx concentrations last from September until May, while night-time $O_3$ was found to reach a minimum during November and December. Previous observations of aerosol concentrations are similarly consistent with our observations: during October and

November 2018, Gunthe et al., (2021) observed aerosol concentrations with a strong nocturnal increase. Gani et al., (2019), in a long-term study, showed that this pattern holds throughout the winter (December – mid-February) and, more weakly, during the spring (February – March). The same study indicates that particulate chloride concentrations are highest during the winter, and extremely low during the summer. We therefore anticipate that conclusions from our own study are likely to be most relevant                    in                    Delhi                    from                    October                    until                    March.

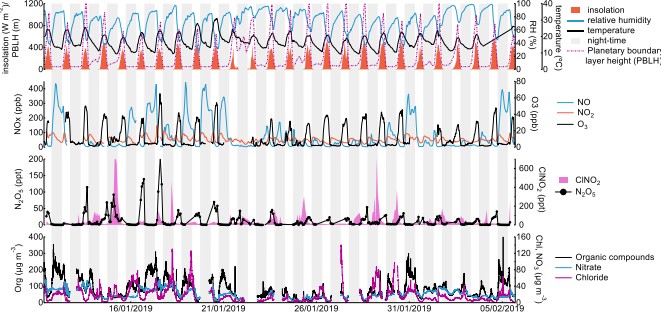

**Figure 2: Full campaign time series of key species and meteorological parameters. Grey and white backgrounds represent night-time and daytime, respectively.**

### 3.2 Unusual diel patterns in Delhi

Figure 3a shows the average diel cycle of $N_2O_5$, as measured by the FIGAERO-CIMS from 11 January to 5 February 2019.

The mean daytime concentration of $N_2O_5$ during the measurement period was 21.9 pptv (standard deviation 29.3 pptv; median 16.0 pptv), compared with a night-time mean of 4.4 (standard deviation 11.3 pptv; median 2.0 pptv; Fig. 3b). A simple 0-dimensional chemical box model was designed to investigate the driving factors behind this inverted $N_2O_5$ pattern in more detail. This simple model successfully recreated the shape of the $N_2O_5$ diurnal pattern observed in the FIGAERO-CIMS dataset (Fig. 3a), and indicated that $NO_3$ is likely to follow a similar diel cycle.

Model results demonstrate that the diurnal $NO_3$ and $N_2O_5$ patterns were influenced most strongly by NO. Our observations show high night-time concentrations of NO, which diminish considerably during the day (Fig. 3c). This pattern has been observed previously in Delhi and is generally attributed to the low, stable nocturnal boundary layer (Stewart et al., 2021) in combination with substantial $NO_x$ sources, such as heavy-duty traffic and biomass burning, being strongly present at night

(Mishra et al., 2023). These night-time NO levels deplete $O_3$ to extremely low concentrations, leaving an excess of NO. Together, these effects ensure that the destruction of $NO_3$ by NO is at its highest during the night, while its production, which requires $O_3$, is at its lowest. In contrast, the presence of $O_3$ and $NO_2$ during the day, coupled with little daytime NO (itself a result of both boundary layer dynamics and reactions with daytime $_{O3}$ and peroxy radicals), results in higher daytime concentrations of $NO_3$ and $N_2O_5$ being sustained than would typically be possible.

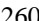

Figure 3: Median diel patterns during the period from 11 Jan – 5 Feb 2019, for the key compounds explored in this study. (a) The median diel cycle of $N_2O_5$ measured by the FIGAERO-CIMS with the interquartile range shaded, alongside the $N_2O_5$ and $NO_3$ simulations from the 0D box model. Error bars indicate a potential 30% error in the estimated calibration constant. (b) Box plots

showing the average $N_2O_5$ concentrations throughout the campaign during the day and the night as measured by the FIGAERO-CIMS, indicating that there is a distinct and consistent decrease in $N_2O_5$ concentrations at night. (c) The observed median diel

patterns of the precursor species NO, NO₂ and O₃ with interquartile ranges shaded. (d) The observed diel pattern of ClNO₂ and particulate chloride throughout the campaign. The error bars here, as in (a), represent a 30% potential calibration error.

Our modelled $N_2O_5$ concentrations are around a two-thirds lower than those observed. Combined with the observed mid-morning peak in $ClNO_2$ (Fig 1d), this is an indication that night-time production of both $N_2O_5$ and $ClNO_2$ continues to some extent in the residual layer, decoupled from the urban canopy layer in which our measurements take place. After sunrise and the ensuing atmospheric instability, the $N_2O_5$ and $ClNO_2$ from the nocturnal residual layer is likely incorporated into the daytime mixed layer and transported by turbulence downwards to the measurement system (Fig. S5). In this study, atmospheric mixing has been quantified using friction velocity ($U^*$) as a scale of the vertical transport of momentum (turbulence). It is presented here as 1-hour means to smooth the stochastic nature of high frequency turbulence and enable the diurnal trend in atmospheric mixing (which occurs over a period of hours) to be viewed more clearly. It can be seen in Fig. S5 that night-time concentrations of $ClNO_2$ are highest with greater atmospheric mixing, supporting the hypothesis that during periods of poor atmospheric mixing the measurement system (which is relatively close to the ground level) is decoupled from air masses above the urban canopy layer that contain higher concentrations of $ClNO_2$. Based on the discrepancy between model and observations for $N_2O_5$ (Fig. 3a) and on the difference between average nocturnal concentrations and the morning peak for $ClNO_2$ (Fig. 3d), we estimate that an average of approximately $18 \pm 3$ pptv of $N_2O_5$ and $12 \pm 3$ pptv of $ClNO_2$ are mixed from the residual layer into the surface layer from above in the mornings, increasing the overall influence of both species during the day.

In addition to model results, oxidation products observed by the FIGAERO-CIMS provide evidence both that Cl is an important oxidant in Delhi, and that oxidation with $NO_3$ takes place during the day. Chloroacetic acid ($C_2H_3O_2Cl$) has previously been designated a tracer compound to indicate the presence of Cl oxidation (Le Breton et al., 2018; Priestley et al., 2018) and was observed here, exhibiting a strong diurnal pattern (Fig. 4b). Similarly, a number of individual compounds that have previously been observed in conjunction with night-time $NO_3$ oxidation of monoterpenes were observed primarily as daytime species during this campaign, with the examples of $C_8H_{11}NO_7$ (Ye et al., 2021; Lee et al., 2016) and $C_{10}H_{15}NO_6$ (Ye et al., 2021; Boyd et al., 2015) displayed here (Fig. 4b). While it is also possible for these CHON compounds to originate from $RO_2 + NO$ reactions during OH-initiated oxidation, the difference in the diurnal cycle here from that observed in other locations indicates a contribution from daytime $NO_3$. The nitrogen-containing organic compounds (CHON) that increased most strongly during the night were all found to be associated with primary biomass burning emissions (Mishra et al., 2023), not oxidation products (Fig. S6). The overall contribution of CHON as a fraction of total organics increased slightly during daylight hours (Fig. 4b), particularly in the gas phase, while results from other parts of the world have typically found the contribution of CHON to increase slightly at night as a result of $NO_3$ oxidation (Ye et al., 2021; Huang et al., 2019). In addition, it is possible for CHON compounds present during the night to have originated from $RO_2 + NO$ reactions during OH-initiated oxidation Daytime peaks in $NO_3$ oxidation products will likely result from both daytime oxidation by $NO_3$ within the surface layer and the mixing down of products from nocturnal oxidation taking place within the elevated residual layer. Given that the other important urban

oxidants, OH and O₃, in addition to Cl, also display daytime maxima, all four key oxidants can be considered daytime species in this environment. As a corollary, there are no significant nocturnal oxidants in the surface layer.

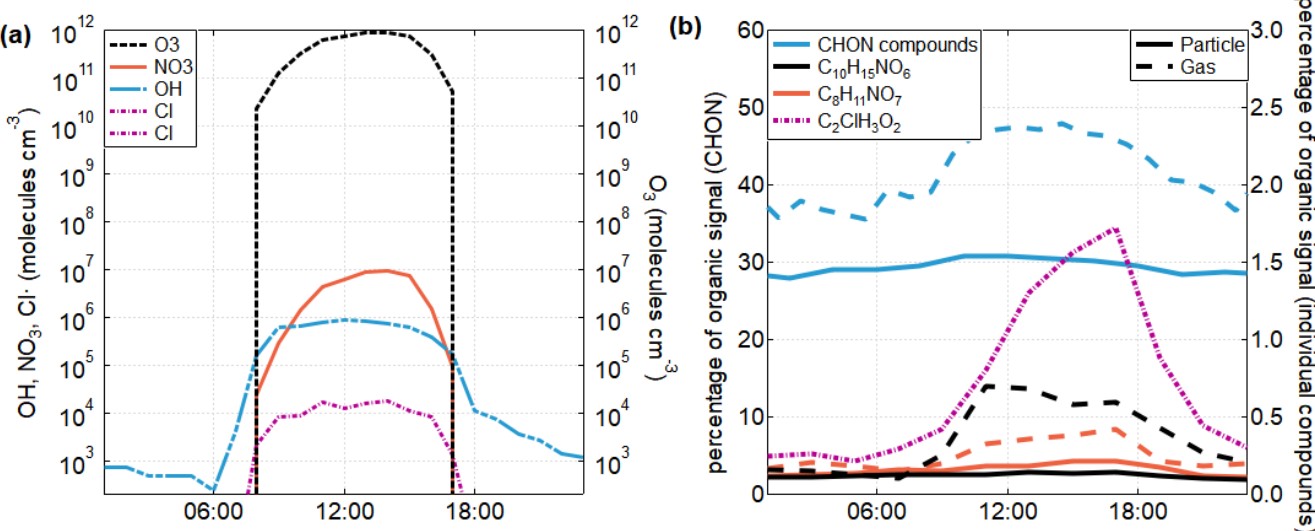

**Figure 4: The diel cycles of key oxidants and oxidation products in Delhi during the study period. (a) The diel cycle of four key oxidants, O₃, Cl, NO₃ and OH. The Cl concentration has been estimated from chlorine-containing species observed by the**
**FIGAERO-CIMS (Methods). Estimated concentrations of the other three oxidants are calculated using the Framework for 0-D Atmospheric Modelling (F0AM) chemical box model (Wolfe et al., 2016) (Methods). (b) The diel cycles of total N-containing organics in the gas and particle phases (blue) with selected compounds C₈H₁₁NO₇ (orange) and C₁₀H₁₅NO₆ (black) also displayed in both the gas and particle phases. These two compounds, both associated with NO₃ oxidation, are most prevalent during the day when the NO₃ concentration peaks, while compounds associated with biomass burning are more prevalent at night (Fig. S6). The diel cycle of**
**C₂H₃O₂Cl, a known tracer of Cl chemistry, is shown in pink. These compounds were not calibrated, so here the percentage contribution of each compound or group of compounds towards the total organic signal is displayed on the y-axes.**

One of the consequences of this shift away from night-time NO₃ within the urban canopy layer is that very little oxidation from any oxidant takes place at the time when sub-micron particulate matter (PM₁) is at its most concentrated; our observations show that PM₁ more than doubles from day to night (Fig. S7a). A previous investigation of the organic aerosol chemical
speciation during this campaign using positive matrix factorisation on aerosol mass spectrometer (AMS) and extractive electrospray ionisation mass spectrometer (EESI) data shows that factors associated with oxidation reactions are enhanced during the day in Delhi, while particulate matter present during the night is the least oxidised (Kumar et al., 2022). The average oxygen to carbon (O:C) ratio of particulate matter during this campaign has been found to be unusually low compared with a global dataset of FIGAERO-CIMS observations in different environments (Huang et al., 2023); the O:C ratio of particulate
matter observed by the CIMS during this campaign was 0.7, compared with values between 0.75 and 0.95 observed elsewhere. If the distribution of oxidants were to change, this would likely increase SOA mass in the region. Such a change could come about if nocturnal NO were to decrease, thereby increasing nocturnal concentrations of NO₃ and N₂O₅ and allowing larger quantities of ClNO₂, and therefore Cl radicals, to be produced.

## 3.3 The potential influence of changes to the diel pattern

Figure 5a shows the relationship between night-time average NO and $N_2O_5$ concentrations in Delhi. It can be seen here that, as NO concentrations during the night decrease, the $N_2O_5$ produced increases substantially. Data from this campaign are displayed alongside observations from Beijing, which were carried out during January 2020. The sampling site for the campaign in Beijing was in the west campus of the Beijing University of Chemical Technology (BUCT). Measurements were taken from the top floor of a five-storey building, at a height of approximately 20 m. The location is comparable to that of the

site in Delhi: it is influenced by local pollution sources including traffic, residential heating and cooking emissions (Cai et al., 2022), and it and can similarly be considered an urban background site. Similar to Delhi, Beijing is a polluted urban centre with a notable contribution from chloride in the particle phase, albeit around 17 times lower than Delhi, on average, based on observations presented here.

The concentrations of $N_2O_5$ in Beijing display a more typical diel pattern, with higher values at night. Night-time $N_2O_5$ regularly exceeds 100 pptv in Beijing, while Delhi's generally remains below 10 pptv. (In this analysis, night-time averages are taken between 20:00 and 05:00 the following morning, in order to prevent residual influence from daytime processes.) The comparison between the two cities allows us to understand more about the impact of Delhi's unusual $NO_x$ cycle on atmospheric chemistry. When the two cities have some overlap in night-time NO concentrations, the quantity of $N_2O_5$ produced at night is

roughly comparable. However, the majority of nights in Beijing have much lower NO concentrations than any observed in Delhi; at these lower concentrations, much more $N_2O_5$ can be produced, in some cases reaching close to 1 ppbv. There is a larger spread in $N_2O_5$ concentrations during nights with low NO, because other factors such as the amount of available $NO_2$ and $O_3$ also have an impact on $N_2O_5$ formation. It is important to note that the lower temperatures in Beijing (where the mean campaign temperature was around 3.4 °C, compared with 16.8 °C in Delhi), will contribute towards the higher $N_2O_5$

concentrations.

There are two key reasons for the substantial difference in the nocturnal NO concentrations between the two cities: first, emissions inventories indicate that vehicular emissions of $NO_x$ decrease by around 90% during the night in Beijing (Jing et al., 2016), compared with a decrease of only around 40% in Delhi (Biswal et al., 2023). Due to traffic regulations restricting the

movement of heavy-duty vehicles during the day, there is increased movement of these vehicles at night in Delhi (Tiwari et al., 2015), leading to greater nocturnal $NO_x$ emissions than can be found in other, comparable cities. Second, the median boundary layer height during the night in winter in Beijing has been reported to be around 100 m (Yang et al., 2020), while data from READY during this campaign (Fig. 2) indicate a nocturnal boundary layer height of 50 m or even lower. As a result, comparable $NO_x$ emissions would result in double the concentration, or even more, in Delhi compared with Beijing. Together,

these factors result in the high nocturnal NO concentrations in Delhi compared with Beijing.

The World Health Organisation (WHO) published new air quality guidelines in 2021 (World Health Organization, 2021), which include guideline 24-hour and annual thresholds of 25 and 10 µg m$^{-3}$ (~13.4 and 5.3 ppbv), respectively for $NO_2$. Our observations show that during the night in Delhi, NO makes up on average 64% of $NO_x$ (NO + $NO_2$) (Fig. S7b). Approximate

night-time NO equivalent thresholds of around 23.8 ppbv in 24 hours or 9.4 ppbv annually can therefore be established. This NO equivalent threshold is displayed in Fig. 5a and can be used to separate the data into two regimes, with nights above the threshold being considered 'high NO' and those below 'low NO'. The majority of nights in Delhi fall within the 'high NO' regime, while the majority of those in Beijing fall into the 'low NO' regime. Within the 'high NO' regime, the average $N_2O_5$ concentration across both datasets is 4.1 pptv, while in the 'low NO' regime, the average $N_2O_5$ concentration per night is 220

pptv. The overlap between NO concentrations in the two cities suggests that it is possible for changes in environmental conditions to shift the atmosphere from one regime to the other: there were some nights during which high NO concentrations in Beijing suppressed the formation of $N_2O_5$, and in Delhi during which low NO concentrations allowed more production of $N_2O_5$. These results suggest that, if night-time NO concentrations were to be reduced in line with WHO guidance, there would likely be a substantial increase in the night-time production of $N_2O_5$.


The relationship between night-time $N_2O_5$ and $ClNO_2$ observations for both Delhi and Beijing can be seen in Fig. 5b. For a given concentration of particulate chloride, larger night-time concentrations of $N_2O_5$ result in the production of more $ClNO_2$, as it increases the number of interactions at the surface of chloride-containing particles. These two datasets together show that, as the ambient concentration of particle-phase chloride increases, this reaction is able to take place more readily and the ratio

of $ClNO_2$ to $N_2O_5$ also increases. This results in much higher quantities of $ClNO_2$ being produced in Delhi, with its higher particulate chloride, for a given concentration of $N_2O_5$ than in Beijing. A linear regression line was fitted to the log of the Delhi dataset to demonstrate this relationship, which has an $r^2$ of 0.58. This pattern indicates that, if night-time $N_2O_5$ concentrations in Delhi were to increase to values as high as those found in Beijing, the high availability of particle-phase chloride could result in $ClNO_2$ production exponentially higher than it is at present. Following the current trendline, an increase in night-time

$N_2O_5$ to 115 pptv – the log-weighted average from the Beijing dataset – would increase the $ClNO_2$ concentrations in Delhi to an average of 1450 pptv, two orders of magnitude higher than the current log-weighted average of 14 pptv.

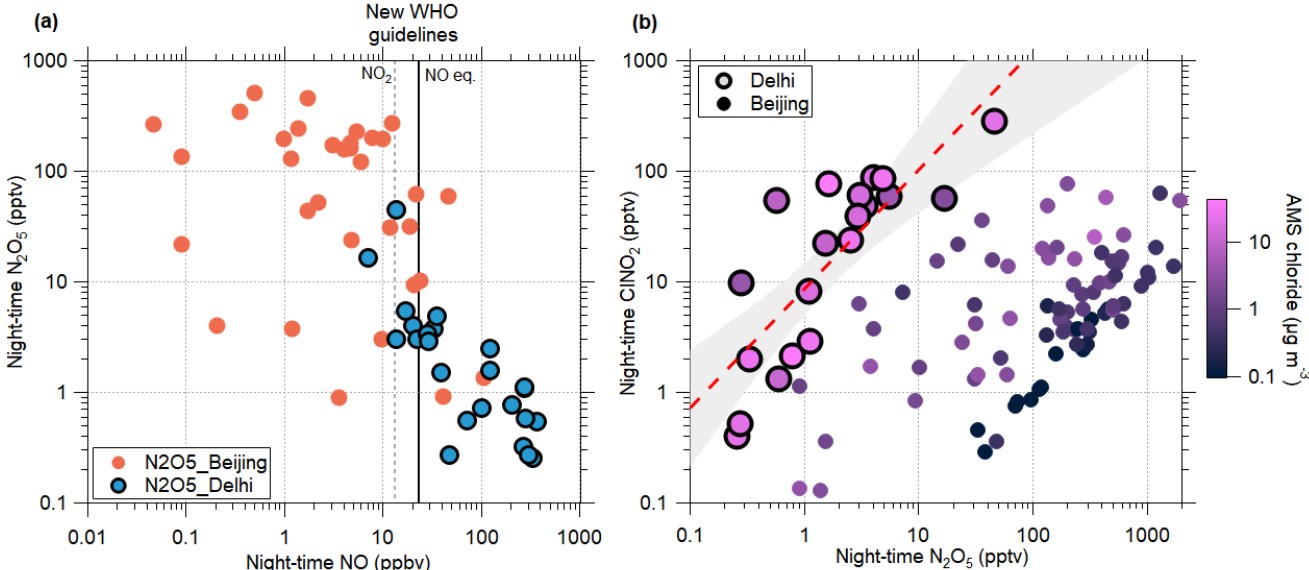

**Figure 5: Relationships between night-time ClNO₂, N₂O₅ and NO during the study period. (A) Average nocturnal concentrations of N₂O₅ plotted against average nocturnal concentrations of NO, for our campaign results from Delhi and from measurements in Beijing from 2020. The new 24-hour WHO guideline for NO₂ concentrations is displayed by the dashed grey line, with the calculated NO equivalent shown by the solid black line. (B) Relationship between nocturnal ClNO₂ and N₂O₅. Colouring shows the average particulate chloride concentration, as measured by the AMS. The red dashed line shows the least-squares regression line for the Delhi dataset, with 95% confidence intervals shown in grey. The equation for the line is given by the equation** $ln(ClNO_2\ (pptv)) = 1.08\ ln(N_2O_5\ (pptv)) + 2.16$**. Data points from between 20:00 and 05:00 the following morning are included.**

We used the F0AM box model (Wolfe et al., 2016; Riedel et al., 2014) to investigate the relative contributions of Cl and NO₃ radicals as oxidants for two common VOCs, alpha-pinene and toluene, under three simulated scenarios. In scenario 1, we used the average diel NO$_x$ concentrations displayed in Fig. 3a as input parameters for the F0AM model. This represents the baseline case. Scenario 2 uses NO$_x$ and O₃ data from a case study day on 15$^{th}$ January 2019. This date was chosen as it includes the night during the campaign that exhibited the lowest night-time NO and highest O₃, leading to the production of a comparatively large amount of ClNO₂, which peaked at 715 pptv. Data were extracted between 18:00 on 14$^{th}$ January and 18:00 on 15$^{th}$ January 2019, replicated and concatenated to form a full diel cycle. This case study represents the most extreme example of the 'low NO' regime observed during our measurement period. Finally, scenario 3 features NO$_x$ and O₃ input parameters that have been taken from the Beijing dataset displayed in Fig. 5 as a stand-in for a more conventional diel cycle in a similarly polluted city. Average sunrise and sunset times differed by only 11 and 13 minutes, respectively, between the two campaigns, which indicates that the timing of concentration changes is likely to be appropriate for the model. The NO and NO₂ maxima and diel patterns are comparable to those observed in Punjab, north of Delhi (Meidan et al., 2022), indicating that they are conceivable for the region. The Beijing dataset was used here because the maximum O₃ concentrations and sunrise and set times are more closely comparable with those in Delhi at this time of year. The concentration of ClNO₂ was fixed at the

beginning of the model run to the highest night-time value in scenario 2, and to 1.45 ppbv in scenario 3, which has been calculated from the fit line in Fig. 5 based on measured $N_2O_5$ concentrations in Beijing. The $ClNO_2$ was then depleted throughout the model run by photolysis, to create oxidising Cl radicals. Scenario 3 is not intended to be a prediction of future atmospheric chemistry in Delhi. Rather, it represents a highly idealised scenario that can be used to explore the potential ramifications of substantial changes to the $NO_x$ and $O_3$ cycle in the context of Delhi's polluted and chlorine-rich environment.


    The results of these model runs are shown in Fig. 6. In scenario 1, which represents the current situation in Delhi, Cl is responsible for 12% of the toluene oxidation, indicating that it is already a major oxidant in the city. No $NO_3$ is produced during the night in this scenario, but as a result of its presence during the day, 25% of the alpha-pinene oxidation is initiated by $NO_3$. During scenario 2, the higher $ClNO_2$ concentration results in almost a doubling of the proportion of toluene oxidised

by Cl to 20%. In addition, the slight increase in night-time $NO_3$ and $O_3$ results in some night-time oxidation of alpha-pinene taking place. This indicates that night-time oxidation is taking place in Delhi, although it is largely restricted to nights that fall within the 'low NO' regime. In this case, $NO_3$ is responsible for 31% of the alpha-pinene oxidation.

    Scenario 3 shows a substantial increase in night-time oxidation from $NO_3$ and daytime oxidation from Cl, due to the large

nocturnal concentration of $O_3$ that is sustained, alongside the lack of NO. Here, the large increase in night-time $NO_3$ production, coupled with the increase in VOC concentrations during the night, leads to $NO_3$ becoming the key oxidant for alpha-pinene, accounting for 82% of its reactivity. Concerning the toluene reactivity, only 16% is initiated by Cl in this case, which is largely due to the associated increase in OH oxidation. In fact, the Cl-initiated toluene reactivity in scenario 3 is almost double that of scenario 2.

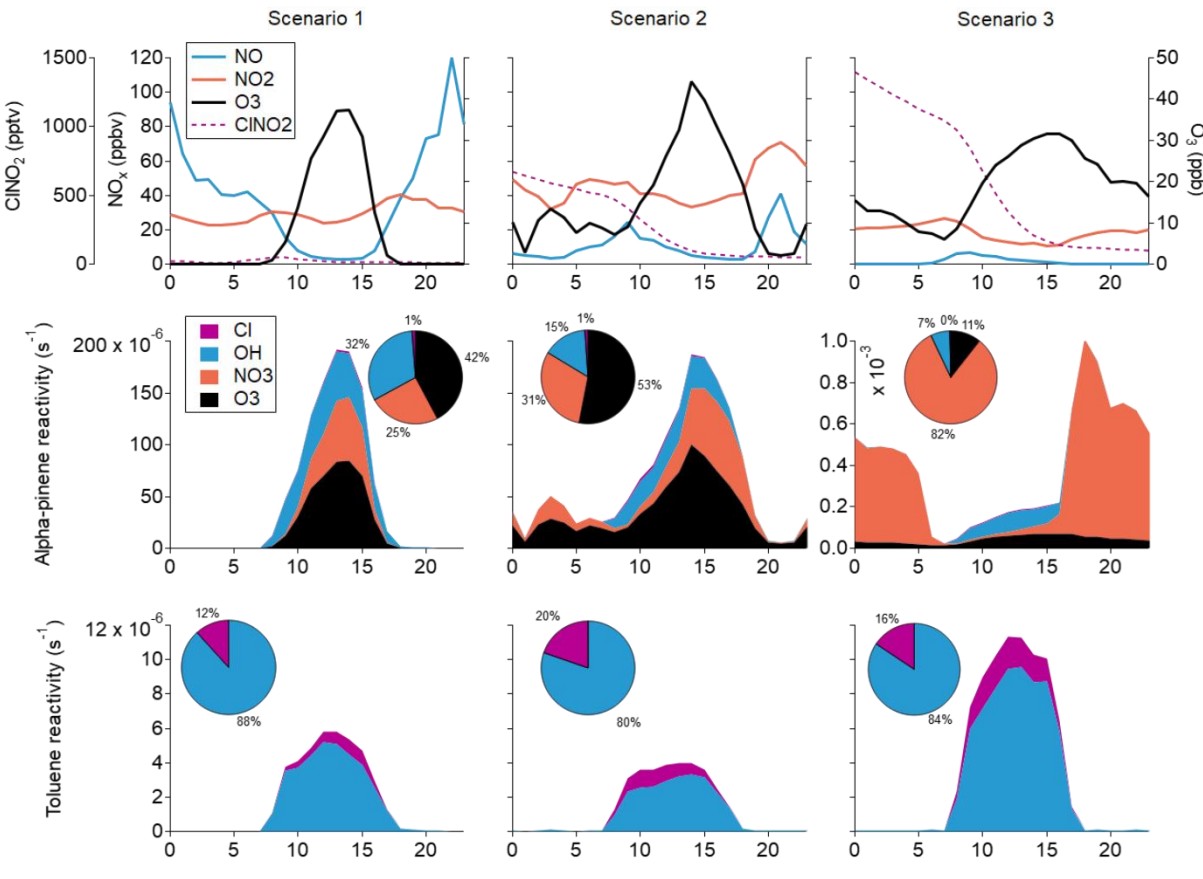


**Figure 6: The reactivity of toluene and alpha-pinene under three simulated scenarios. (A), (B) and (C) show the NO, NO2, O3 and ClNO2 concentrations for scenarios 1, 2 and 3, respectively. In all three cases, NO and NO2 values in the model are held to the measurements, as is O3 in scenarios 2 and 3. For ClNO2, the initial value is fixed and evolves throughout the model run for scenarios 2 and 3. (D), (E) and (F) indicate the reactivity of alpha-pinene in scenarios 1, 2 and 3 respectively, and (G), (H) and (I) the reactivity of toluene.**


## 4. Conclusions

The analysis presented here demonstrates that the diel cycles of the typically nocturnal species $NO_3$ and $N_2O_5$ are inverted during the winter in Delhi's surface layer. This is due to the presence of large concentrations of NO during the night; which is the result of the night-time compression of the boundary layer, coupled with large night-time emissions of $NO_x$. This nocturnal

NO depletes $NO_3$ during the night and therefore results in both $NO_3$ and $N_2O_5$ consistently peaking during the day. This unique diel chemical pattern limits the formation of night-time $ClNO_2$. Despite this constraint on $ClNO_2$ formation, the extremely high particulate chloride loadings available in Delhi still result in enough Cl production for it to be an important atmospheric oxidant. Model simulations carried out here using the F0AM model demonstrate that the role of Cl as an oxidant of toluene could be more than 10 times higher if the diel cycle of $NO_x$ were more representative of that observed in comparable urban

atmospheres such as Beijing. Similarly, the role of $NO_3$ as an oxidant of alpha-pinene was shown to be around 30 times lower in Delhi than it would be if the $NO_x$ pattern were more typical. This is a particularly significant difference from the majority of urban centres, as $NO_3$ would otherwise be the only oxidant that is most prominent during the night, when the concentration of particulate matter is at its highest.

A number of clean air policies, including the latest update to the WHO guidelines on ambient air pollution (World Health Organization, 2021), emphasise the control of $NO_x$ pollution in urban environments. This study highlights the complexity of atmospheric chemistry in a highly-polluted urban environment and indicates that it is important to monitor the impacts of changes to specific species. Given the delicate chemical balance in Delhi's winter atmosphere, there is potential for significant changes in night-time NO pollution to result in increased production of $NO_3$ and $N_2O_5$ in the surface layer. Such a change

could be initiated by, for example, a reduction in nocturnal vehicle emissions. Our results indicate that both $NO_3$- and Cl-initiated oxidation could be substantially increased if the diurnal cycle for $NO_x$ in Delhi were more representative of that seen in other major cities; this would increase the formation of SOA and therefore worsen urban haze. We strongly recommend careful monitoring of $NO_x$, $O_3$ and particulate chloride levels in Delhi, as there is potential for disruption to the $NO_x$ cycle. It is important for more research to be undertaken into this problem in order to understand the potential ramifications in more

depth. Given the importance of boundary layer dynamics for pollutant concentrations, and the potential influence of $N_2O_5$ and $ClNO_2$ formation in the residual layer, we would recommend that future projects in Delhi incorporate a vertical component. This could include measurements being made at different heights or vertical box-modelling, which would contribute substantially towards a more complete understanding of the processes explored here.

**Data availability.** Data from this paper will be made available on the Bolin Centre Database.

**Author contributions.** The concept for this manuscript was established by SLH and CM, with input from DMB, JT, UB, ASHP and SNT. Investigation and measurements were carried out by SLH, VK, DSW, SM, NR, AS, DG, FZ, YL, WN, YL, WM, CY, KRD and CM. MK, UB, ASHP, SNT and CM acquired funding for this project. The writing was carried out by SLH. Reviewing and editing was undertaken with input from all co-authors.

**Acknowledgements.** The research conducted in this paper was supported by the Knut and Alice Wallenberg Foundation (project CLOUDFORM, grant no. 2017.0165). One of the corresponding authors (SNT) gratefully acknolwedges the financial support provided by the Central Pollution Control Board (CPCB), Government of India, to conduct this research under grant no. AQM/Source apportionment_EPC Project/2017. We also acknowledge funding from the SDC Clean Air Project in India (grant no. 7F-10093.01.04), the Swiss National Science Foundation (2000021_169787). KRD acknowledges support by the

Schweizerischer Nationalfonds (SNF) mobility grant (grant no. P2EZP2_181599) and Ambizione grant (grant no. PZPGP2_201992).

**Competing interests.** The authors declare that they have no conflict of interest.

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
