# Peer review of "Night-time NO emissions strongly suppress chlorine and nitrate radical formation during the winter in Delhi"

_EGUsphere, 2023_

## Referee Comment (RC2)

**Review of "Night-time NO emissions strongly suppress chlorine and nitrate radical formation during the winter in Delhi"**

Overall comment:
This manuscript analyzes ground based CIMS field measurements in Delhi during the winter. In combination with a 0D box model, the authors assess how high NO surface concentrations at night impact the radical budget in Delhi. Insights also come from an interesting comparison of their measurements in Delhi to their measurements in Beijing, a city with lower nighttime NO surface concentrations.

The paper presents interesting conclusions about the unique chemical environment in Delhi and discusses implications for how the current chemistry should inform further emissions reductions. Overall, the content of the paper will be of interest to the atmospheric chemistry community, and I recommend publication after addressing the concerns listed below.

General remarks:
There is a lack of clarity throughout the manuscript distinguishing the effects of boundary layer dynamics and the effects of chemistry on observed concentrations. For example, the high NO surface concentrations observed at night are largely attributed to high nocturnal NO emissions but without any quantitative justification for this attribution. Though large NO emissions at night can be a big contributor to high nighttime NO concentrations at the surface, the height of the nocturnal surface layer also exerts control on nighttime NO concentrations. This is important to address, especially when assessing the differences between Delhi and Beijing and when assessing the impacts of possible emissions controls. One could imagine a scenario in which the nocturnal boundary layer dynamics are dramatically different between the two cities, meaning that the differences in chemistry observed would be driven largely by dynamics rather than by emissions. This is an important aspect to address, at the very least through a simple comparison of NO emissions inventories between the two cities and perhaps also with an assessment of what meteorological/dynamical conditions are coincident with particularly high- and low-NO surface concentrations at night.

As the authors hint at several times, nighttime $NO_3$ chemistry is generally understood to be most important in the nocturnal residual layer, decoupled from the fresh NO emissions at the surface. The products of the chemistry in the residual layer can then impact concentrations at the surface when sunlight-driven convection begins in the morning. However, the 0D box model used in this work does not account for any residual layer chemistry. The authors attest that residual layer chemistry is likely important, concluding that differences between their model and their measurements are likely driven by mixing from the residual layer. Given the importance of the residual layer for nighttime $NO_3$ chemistry, it seems remiss to not include some sort of accounting of residual layer chemistry in the box model, especially since the box model is then used to assess overall oxidant budgets. Something as simple as a 1D 2-box model (one box for the surface layer, one box for the residual layer, with mixing between boxes) could address this. I also recommend including some discussion of the overall structure of the nocturnal boundary layer (including distinguishing the surface vs residual layer) in the introduction when introducing nocturnal chemistry.

Specific comments:
Line 48: is the 10% by mass?

Line 160: R6 and R7 are only defined in the supplement, so the reference here is confusing.

Line 162: "outlined in more detail below" should be changed to "outlined in more detail in the supplement"

Line 163: should reference Eq 1

Lines 160-165: How are you parameterizing the $ClNO_2$ yield? I could only find details about $N_2O_5$ uptake, but not on the yield of $ClNO_2$ from $N_2O_5$ hydrolysis.

Lines 176-177: Can you give some sort of justification for whether these select VOCs are representative of total VOC reactivity?

Lines 211-212: Boundary layer dynamics can also play a role in the high NO concentrations, in addition to the emissions and low $O_3$ concentrations mentioned here!

Line 215: Here the high OA concentrations at night are attributed to dynamics. The effect of dynamics can affect NO and $O_3$ concentrations as well!

Lines 231-232: Are NO emissions in Delhi dramatically different between day and night? Or are the NO concentration differences observed between day and night primarily a result of dynamics?

Lines 232-235: This sentence is really important in acknowledging the effects of both dynamics and emissions on observed concentrations. I recommend making sure this idea permeates throughout the manuscript.

Line 237: Is the "little daytime NO" a result of differences in emissions or differences in dynamics?

Line 235-238: Consider adding a plot in the supplement of the relative importance of different $NO_3$ loss pathways over the diel cycle.

Figure 2: I think there is a typo in the legend to the right of the top panel ("daynight" should be "nighttime").

Lines 248-259: This acknowledgment of the importance of residual layer chemistry is really important—I think it should be included somehow in your box model, perhaps through the use of a 1D 2-box model.

Line 253: Can you include a little more description of how you quantify "atmospheric mixing," including what time intervals it is calculated over?

Line 257: Can you use these estimates to figure out something about the mixing timescales and residual layer concentrations of $N_2O_5$, $ClNO_2$, etc.?

Lines 267-272: I think it's important to note that CHON compounds can also be derived from $RO_2 + NO$ reactions during OH-initiated oxidation.

Line 288: Can you provide some quantitative justification (i.e., data) for saying that the aerosol oxidation state in Delhi is "very low"?

Line 293: Can you be more quantitative here (rather than just "unusually low")?

Line 300: Can you include a few other important details about the Beijing measurement site (was it also ground-based, was it also at a background urban site)?

Line 303: I think a comparison of NO emissions inventories between Delhi and Beijing is an important part of this comparison to help distinguish whether observed differences are due more to dynamics versus due more to emissions.

Line 305: Replace "where" with "when."

Line 309: Because $N_2O_5$ is also thermally stabilized, a comparison of temperatures between Delhi and Beijing is another important factor to consider here.

Line 325-326: I suggest adding the following italicized phrase for clarity: "*At a given pCl concentration,* larger nighttime concentrations of $N_2O_5$ result…"

Figure 6: Why are constrains on $O_3$ and initial $ClNO_2$ different between scenario 1 and scenarios 2 and 3?

Line 387: Are these large concentrations of NO a result of emissions or dynamics? This has important implications for the conclusions you later draw about emissions reductions, etc.

---

## Author Comment (AC1)

**Review of "Night-time NO emissions strongly suppress chlorine and nitrate radical formation during the winter in Delhi"**

This manuscript presents an atmospheric chemistry story from 25 days of data (11 January – 5 February) from a FIGAERO-CIMS complemented with AMS, NOx, and O3 measurements in Delhi, India. The authors use these measurements with a 0-D chemical box model to understand N2O5 sources and sinks. The study finds that the high night-time NO depletes O3, NO3, and N2O5 (a precursor to ClNO2 and thus Cl). The authors finally suggest that decreasing NO emissions will result in NO3 becoming an important nocturnal oxidant during the night and Cl during the day. The manuscript provides interesting insights into the "unusual" atmospheric chemistry, especially the role of various atmospheric oxidants, in Delhi which may be relevant for other polluted cities in the Indo-Gangetic Plain where there is similar combination of large emissions (including NOx) and unfavorable (nighttime) meteorology.

We thank the reviewer for these comments, which have improved our manuscript. Please find our responses below in blue font. Extracts from the manuscript are presented in pink, with changes from the original underlined.

Some comments:

1. The authors should be careful in the distinction between "emissions" and "levels". These are not always interchangeable. As the authors themselves discuss in the excellent discussion in the last paragraph of page-9, boundary layer dynamics (unfavorable nighttime meteorology) plays an important role in Delhi. As such, it may not be simply the "nighttime NO emissions", but the "nighttime NO levels" that are of relevant to the discussion. While this may seem trivial, this is extremely relevant from a policy perspective and specifically how the authors discuss policy implications. I suggest that the authors carefully go through the manuscript and check if the use of "emissions" and "levels" is correct and intentional.

We agree with the reviewer that this detail is important and thank them for highlighting it. We have made changes in the manuscript in the following places:

Line 256: 'These night-time NO levels deplete O$_3$ to extremely low concentrations…'

Line 372: 'These results suggest that, if night-time NO concentrations were to be reduced in line with WHO guidance…'

We have chosen to retain the word 'emissions' in the manuscript title. Although we appreciate that boundary layer dynamics will contribute towards the high concentrations, we still feel it is important to highlight the importance of the high night-time emissions.

Our second reviewer also suggested that our original manuscript did not sufficiently explore the role of boundary layer dynamics on the night-time NO concentration in Delhi. As such, we have made a number of changes to the manuscript in order to give this phenomenon sufficient weight. Details of these changes can be found in our responses to reviewer 2.

2. Delhi has large seasonal variations in pollution loadings (including aerosol and gas) because of a combination of changing sources (especially heating in winter) and meteorology. As a result, a 25-day study from January (2019) should be careful in generalizing the loadings (including chloride) and chemistry for the entire year. I am not saying that the findings are not important, but that they should be put in context of the study period. For example, the authors could use previously published year-round data for aerosol composition and NOx (even if no FIGAERO-CIMS) to put the study period in context.

We agree with the reviewer that the larger-scale annual context is extremely important for a study such as this. We have been careful to note in the title that we are discussing a wintertime phenomenon, as we feel that this pattern is unlikely to hold true during the summer.

We have now included a more explicit literature analysis of NOx, $O_3$ and particulate matter concentrations and patterns throughout the year, to seat our own results in context. The following text has been added to Section 3.1:

Lines 230-39: 'Similarly high concentrations of NO and low concentrations of $O_3$ at night have been observed in previous wintertime studies in Delhi. For example, Nelson et al., (2021, 2023) observed a very similar diel NO and $O_3$ pattern to that displayed here in October-November 2018. In a year-long study, Sharma et al., (2021) demonstrated that these high night-time NOx concentrations last from September until May, while night-time $O_3$ was found to reach a minimum during November and December. Previous observations of aerosol concentrations are similarly consistent with our observations: during October and November 2018, Gunthe et al., (2021) observed aerosol concentrations with a strong nocturnal increase. Gani et al., (2019), in a long-term study, showed that this pattern holds throughout the winter (December – mid-February) and, more weakly, during the spring (February – March). The same study indicates that particulate chloride concentrations are highest during the winter, and extremely low during the summer. We therefore anticipate that conclusions from our own study are likely to be most relevant in Delhi from October until March.'

3. The authors should also include the study-period in the figure captions. For example, instead of "The diel cycles of key oxidants and oxidation products in Delhi" it should be "The diel cycles of key oxidants and oxidation products in Delhi during the study period" (even better if you include the study dates)

The captions have been updated for Figs. 3, 4 and 5.

4. How are "night" and "day" defined for the analysis? I could not find this in the methods.

In Sections 3.1 and 3.2, where the overall campaign results are discussed, we used the average sunrise and sunset times to split the dataset into 'daytime' and 'night-time' values. We have now added the following text to clarify this:

Lines 215-6: 'For these calculations, an average campaign sunrise time of 07:12 and sunset of 17:53 were used to split the data into daytime and night-time values.'

However, for the average night-time datapoints presented in Fig. 5, we include a buffer of 2 hrs at each side of the night to ensure there are no residual daytime influences on the average. In the caption for Fig. 5, we have already included the following text:

Line 393: 'Data points from between 20:00 and 05:00 the following morning are included.'

We now feel this is also important to highlight in the main text, and have therefore included the following:

Lines 339-41: '(In this analysis, night-time averages are taken between 20:00 and 05:00 the following morning, in order to prevent residual influence from daytime processes.)'

5. In Section 2.1, the name of the centre is incorrectly written. I believe the correct name of the centre is the same as the affiliation of one of the authors of the study. Also, are the authors sure that the distance from the major roadway is just 80m? Please also include which floor the measurements were conducted in.

We thank the reviewer for spotting this. We have now corrected the name of the centre in line 79.

The IIT-Delhi campus lies on the outer ring-road (Gamal Abdel Nasser Marg, here), so it is immediately north of the measurement site – about 80 m according to Google maps. The slightly larger inner ring-road (Mahatma Gandhi Rd) is 2.4 km to the north. We have chosen to reference the slightly smaller, outer ring-road, here, which is close to the measurement site. It is still a large, busy road and, due to its proximity, we consider that it is more likely to have direct impacts on our measurements in this case.

Measurements were conducted in the fourth floor; this is already stated in line 81.

6. In Section 2.3, line 151, the measurement location for meteorological data should be included.

Meteorological parameters were supplied from the measurement station at the airport, 8 km to the west of the measurement site. This information has now been included in lines 153-4.

7. In Figure 2, adding MLH from reanalysis dataset such as ERA5 or MERRA2 may provide interesting insights.

The planetary layer boundary height has now been included in Fig. 2, and a description of the origin of the data added to the method section.

Lines 186-8: 'The planetary boundary layer height (PBLH) displayed in Fig. 2 and the friction velocity ($U^*$) displayed in Fig. S5 were obtained from the Real-time Environmental Applications and Display sYstem (READY; Rolph et al., 2017) website, and were available at 3 h resolution.'

Thanks to the authors for writing an interesting atmospheric chemistry manuscript. I hope that the comments above help improve the article.

References

Gani et al., *Atmos. Chem. Phys.,* 19, 6843-6859, https://doi.org/10.5194/acp-19-6843-2019, 2019.

Gunthe et al., *Nature Geosci.,* 14, 77-84, https://doi.org/10.1038/s41561-020-00677-x, 2021.

Nelson et al., *Atmos. Chem. Phys.,* 21, 17, 13609-13630, https://doi.org/10.5194/acp-21-13609-2021, 2021.

Nelson et al., *Environ. Sci. Technol. Lett.,* https://doi.org/10.1021/acs.estlett.3c00171, 2023.

Sharma et al., *Urban Clim.,* 39, 100980, https://doi.org/10.1016/j.uclim.2021.100980, 2021.

---

## Author Comment (AC2)

**Review of "Night-time NO emissions strongly suppress chlorine and nitrate radical formation during the winter in Delhi"**

Overall comment:
This manuscript analyzes ground based CIMS field measurements in Delhi during the winter. In combination with a 0D box model, the authors assess how high NO surface concentrations at night impact the radical budget in Delhi. Insights also come from an interesting comparison of their measurements in Delhi to their measurements in Beijing, a city with lower nighttime NO surface concentrations.

The paper presents interesting conclusions about the unique chemical environment in Delhi and discusses implications for how the current chemistry should inform further emissions reductions. Overall, the content of the paper will be of interest to the atmospheric chemistry community, and I recommend publication after addressing the concerns listed below.

Thank you for this constructive review; responding to these comments has improved our manuscript. Our responses below are in blue, and extract from the new manuscript are presented in pink, with changes from the original underlined.

General remarks:
There is a lack of clarity throughout the manuscript distinguishing the effects of boundary layer dynamics and the effects of chemistry on observed concentrations. For example, the high NO surface concentrations observed at night are largely attributed to high nocturnal NO emissions but without any quantitative justification for this attribution. Though large NO emissions at night can be a big contributor to high nighttime NO concentrations at the surface, the height of the nocturnal surface layer also exerts control on nighttime NO concentrations. This is important to address, especially when assessing the differences between Delhi and Beijing and when assessing the impacts of possible emissions controls. One could imagine a scenario in which the nocturnal boundary layer dynamics are dramatically different between the two cities, meaning that the differences in chemistry observed would be driven largely by dynamics rather than by emissions. This is an important aspect to address, at the very least through a simple comparison of NO emissions inventories between the two cities and perhaps also with an assessment of what meteorological/dynamical conditions are coincident with particularly high- and low-NO surface concentrations at night.

We agree that the height of the nocturnal surface layer also exerts control on night-time NO concentrations, and on the reviewer's suggestion, we have now emphasised this more strongly throughout the manuscript. Detailed responses and changes to the manuscript are highlighted below in response to specific questions raised by the reviewer.

We maintain that, although boundary layer dynamics will play an important role, night-time $NO_x$ emissions also contribute substantially towards the observed NO patterns. A night-time increase

in NO with the highest concentrations between 10 pm and 2 am (Fig. 3), with average night-time NO concentrations that are 10 times higher than in Beijing (Fig. 5) cannot be explained by boundary layer dynamics alone. This would also require elevated production during the night that exceeds the sink by ozone destruction.

Traffic has been shown to be the dominant source of $NO_x$ in Delhi, producing 66-74% of emissions (Gulia et al., 2015), and traffic regulations in Delhi restrict the movement of the most polluting vehicles (eg heavy-duty vehicles) during the day, which results in large amounts of traffic movement during the night. Approximately 80,000 heavy-duty vehicles are estimated to enter Delhi each night, leading to large-scale night-time emissions of $NO_x$ (Tyagi et al., 2016). In contrast, the highest NO concentration in Beijing is found during the morning rush hour, with a peak at 8 am (see eg Akimoto et al., 2019).

According to a recent emissions inventory for Delhi (Biswal et al., 2023), vehicular $NO_x$ emissions in Delhi decrease by around 40% at night compared with their daytime peak. This is less dramatic than the nocturnal decrease in emissions of organic matter, VOCs and CO (for which night-time emissions are 70% - 90% lower than daytime). This contrast was attributed to emissions from heavy-duty vehicles during the night. A similar study of vehicular emissions in Beijing showed a decrease of around 90% in $NO_x$ emissions during the night compared with the daytime peak (Jing et al., 2016). This indicates that it is not only boundary layer dynamics, but also higher emissions that contribute towards Delhi's high nocturnal NO.

During the winter, the urban boundary layer height in Beijing is around 100 m at night (Yang et al., 2020), compared with a height during this campaign in Delhi of around 50 m, or perhaps slightly lower (now displayed in Fig. 2). Previous literature suggests an average night-time boundary layer height in the winter in Delhi of 100 m, similar to that reported for Beijing (Raj et al., 2021). We would therefore expect boundary layer dynamics to account for up to a factor of 2-4 difference in the concentrations of NO between Delhi and Beijing (the higher value would be if the boundary layer in Delhi reduces to as low as 25 m, which would be difficult to measure accurately). The remainder of the difference is likely the result of differences in night-time emissions.

Several previous studies have observed and commented on the same pattern in Delhi, and concluded that these night-time NO concentrations can only be explained by a combination of both high night-time emissions and a lower night-time boundary layer (eg Tiwari et al., 2015; Nelson et al., 2023). We agree with this assessment. We have updated the manuscript in several places (outlined in more detail below) to emphasise the importance of boundary layer dynamics for night-time NO concentrations.

As the authors hint at several times, nighttime $NO_3$ chemistry is generally understood to be most important in the nocturnal residual layer, decoupled from the fresh NO emissions at the surface. The products of the chemistry in the residual layer can then impact concentrations at the surface

when sunlight-driven convection begins in the morning. However, the 0D box model used in this work does not account for any residual layer chemistry. The authors attest that residual layer chemistry is likely important, concluding that differences between their model and their measurements are likely driven by mixing from the residual layer. Given the importance of the residual layer for nighttime $NO_3$ chemistry, it seems remiss to not include some sort of accounting of residual layer chemistry in the box model, especially since the box model is then used to assess overall oxidant budgets. Something as simple as a 1D 2-box model (one box for the surface layer, one box for the residual layer, with mixing between boxes) could address this. I also recommend including some discussion of the overall structure of the nocturnal boundary layer (including distinguishing the surface vs residual layer) in the introduction when introducing nocturnal chemistry.

We agree that boundary layer dynamics are important, and that a 2-box model would be an interesting way to investigate this. We did in fact attempt something like this during the preparation of the manuscript. However, it quickly became clear that we did not have sufficient information to justify any assumption on the amount of mixing between the two layers, both in the evening (how much $N_2O_5$ etc ends up in the residual layer?) and in the morning (how much mixes back down again and to what extent does it influence our observed concentrations?)

As such, we concluded that such a model falls beyond the scope of what is possible in this paper. It would require more data on meteorological parameters as a function of altitude, as well as, ideally, information on the vertical distribution of pollutants, in order to constrain the approach. We have therefore not implemented such a model in this paper. However, it would be interesting to explore these dynamics in more detail for a future project. We have included this recommendation for future research in the manuscript:

Lines 459-62: 'Given the importance of boundary layer dynamics for pollutant concentrations, and the potential influence of $N_2O_5$ and $ClNO_2$ formation in the residual layer, we would recommend that future projects in Delhi incorporate a vertical component. This could include measurements being made at different heights or vertical box-modelling, which would contribute substantially towards a more complete understanding of the processes explored here.'

Specific comments:
Line 48: is the 10% by mass?

This 10% value is by mass. We have now added this clarification into the manuscript at line 48.

Line 160: R6 and R7 are only defined in the supplement, so the reference here is confusing.

The phrasing has now been altered to reduce confusion.

Line 162-3: 'The heterogeneous reaction rates for N$_2$O$_5$ and NO$_3$ at particle surfaces (R6 and R7 in the supplement) rely on $c_{N2O5}$ and $c_{NO3}$, the average molecular speed of an N$_2$O$_5$ or NO$_3$ molecule ($\bar{c}_X$) …"

Line 162: "outlined in more detail below" should be changed to "outlined in more detail in the supplement"

This has now been updated.

Line 163: should reference Eq 1

This has been updated in the text.

Lines 160-165: How are you parameterizing the ClNO$_2$ yield? I could only find details about N$_2$O$_5$ uptake, but not on the yield of ClNO$_2$ from N$_2$O$_5$ hydrolysis.

The simple box model was used to investigate the formation of N$_2$O$_5$, so there was no parameter for the ClNO$_2$ yield. Heterogeneous ClNO$_2$ formation was not modelled in this study. The ClNO$_2$ concentrations used later in the manuscript are from our measurements.

Lines 176-177: Can you give some sort of justification for whether these select VOCs are representative of total VOC reactivity?

We thank the reviewer for raising this. We also note that the figure was, in fact, updated prior to submission to include a slightly broader range of VOCs, but we neglected to update the text to reflect this. The model in fact also includes the 5 most abundant VOCs measured by a C2-C6 analyser during the campaign, in addition to those already listed in the text. We have now updated this in the text (see below).

We do not have measurements for some of the more abundant VOCs that are likely to be present in Delhi; most notably, methanol and acetone (Tripathi et al., 2022). For this reason, our calculation of the N$_2$O$_5$ concentration is likely to represent a slight over-estimate. Nevertheless, we note that the VOC sink for NO$_3$ was minimal compared with the other sinks (namely, photolysis and the reaction with NO). It is therefore likely that our under-estimate of the total VOC concentration will have a relatively small impact.

We have updated the manuscript to reflect this.

Lines 144-6: 'A proton-transfer-reaction time-of-flight mass spectrometer (PTR, Ionicon Analytical G.m.b.H, Innsbruck, Austria) was used to measure the larger VOCs, and smaller VOCs were observed using an AirmoVOC C2-C6 analyser, model A22022 (Chromatotec ®, France).'

Lines 179-80: 'The chemical composition of VOCs included in the model were $C_2H_6$, $C_2H_4$, $C_3H_8$, $C_4H_{10}$, $C_6H_{14}$, $C_6H_6$, $C_6H_7$, $C_8H_{10}$, $C_9H_{12}$, $C_5H_8$ and $C_{10}H_{16}$.'

Lines 181-4: 'Measurements were not available for some of the more abundant VOCs in Delhi such as methanol and acetone (Tripathi et al., 2022), and as such, the magnitude of the VOC sink is likely an underestimate. Nevertheless, this sink was minimal compared with the $NO_3$ loss via interaction with NO and photolysis. This underestimate is therefore unlikely to result in substantial impact on the estimated $N_2O_5$ concentration.'

Lines 211-212: Boundary layer dynamics can also play a role in the high NO concentrations, in addition to the emissions and low $O_3$ concentrations mentioned here!

We thank the reviewer for highlighting this and we have now updated the text as follows:

Lines 220-2: 'This is an unusual feature and is likely a consequence of more heavy-duty vehicles using the roads at night due to daytime restrictions (Tobler et al., 2020), combined with the low night-time boundary layer height (Raj et al., 2021) and low nocturnal $O_3$ levels limiting the conversion of NO to $NO_2$.'

Line 215: Here the high OA concentrations at night are attributed to dynamics. The effect of dynamics can affect NO and $O_3$ concentrations as well!

We have now aimed to acknowledge this more thoroughly throughout the manuscript.

Lines 231-232: Are NO emissions in Delhi dramatically different between day and night? Or are the NO concentration differences observed between day and night primarily a result of dynamics?

Emissions of $NO_x$ from traffic are somewhat lower during the night, by approximately 40% compared with the daytime, according to Biswal et al. (2023). However, night-time $NO_x$ emissions from traffic are still unusually large in Delhi compared with many other cities, primarily due to the prevalence of heavy-duty vehicles at night. The night-time increase in NO concentration is therefore a result of both the boundary layer compression and the continued emission of $NO_x$ throughout the night. During the day, NO concentrations fall to zero, due to a combination of both the expanding boundary layer and the impact of increased daytime $O_3$.

Nevertheless, we maintain that the high night-time production of NO plays an important role in this pattern. We therefore feel that it is important to retain the current reference about night-time emissions, alongside the existing discussion about the impact of boundary layer processes.

We have broadened our discussion of the differences in NOx emissions in lines 350-58, which are written out in full later in this response.

Lines 232-235: This sentence is really important in acknowledging the effects of both dynamics and emissions on observed concentrations. I recommend making sure this idea permeates throughout the manuscript.

We have now aimed to increase discussion of the impact of boundary layer effects throughout the manuscript, as is detailed in several other responses here.

Line 237: Is the "little daytime NO" a result of differences in emissions or differences in dynamics?

This is the result of boundary layer expansion during the day, in combination with the remaining NO reacting with $O_3$ and peroxy radicals to form $NO_2$. We have now included this in the main text:

Line 258-60: 'In contrast, the presence of $O_3$ and $NO_2$ during the day, coupled with little daytime NO (itself a result of both boundary layer dynamics and reactions with daytime $O_3$ and peroxy radicals), results in higher daytime concentrations of $NO_3$ and $N_2O_5$ being sustained than would typically be possible.'

Line 235-238: Consider adding a plot in the supplement of the relative importance of different $NO_3$ loss pathways over the diel cycle.

This plot has now been added as Fig. S3. The majority of $NO_3$ loss is due to interaction with NO throughout the full diel cycle.

Figure 2: I think there is a typo in the legend to the right of the top panel ("daynight" should be "nighttime").

This has now been corrected in the manuscript.

Lines 248-259: This acknowledgment of the importance of residual layer chemistry is really important—I think it should be included somehow in your box model, perhaps through the use of a 1D 2-box model.

We agree that this would be an ideal way to approach this problem. However, we were unable to collect enough information to be able to parameterise such a model sufficiently. A 2-box model is therefore unfortunately outside the scope of this study. We would be interested in looking into this in a future study with additional measurements. In lines 458-61, we have recommended this approach for future studies.

'Given the importance of boundary layer dynamics for pollutant concentrations, and the potential influence of $N_2O_5$ and $ClNO_2$ formation in the residual layer, we would recommend that future projects in Delhi incorporate a vertical component. This could include measurements being made

at different heights or vertical box-modelling, which would contribute substantially towards a more complete understanding of the processes explored here.'

Line 253: Can you include a little more description of how you quantify "atmospheric mixing," including what time intervals it is calculated over?

Atmospheric mixing here has been quantified using the friction velocity ($U^*$) as an indication of turbulence. Values were obtained from the Real-time Environment Applications and Display sYstem (READY; Rolph et al., 2017) website, and were calculated over 3-hour intervals. More information on this has been added to both the method and results sections, as follows:

Lines 186-188: 'The planetary boundary layer height (PBLH) displayed in Fig. 2 and the friction velocity ($U^*$) were obtained from the Real-time Environmental Applications and Display sYstem (READY; Rolph et al., 2017) website, and was available at 3 h resolution.'

Lines 274-7: 'In this study, atmospheric mixing has been quantified using friction velocity ($U^*$) as a scale of the vertical transport of momentum (turbulence). It is presented here as 3-hour means to smooth the stochastic nature of high frequency turbulence and enables the diurnal trend in atmospheric mixing (which occurs over a period of hours) to be viewed more clearly.'

Line 257: Can you use these estimates to figure out something about the mixing timescales and residual layer concentrations of $N_2O_5$, $ClNO_2$, etc.?

We do not have enough data to quantify the atmospheric mixing of specific compounds. In future experiments, tethered balloon measurements of meteorological and pollutant concentrations would be ideal to quantify atmospheric mixing and its impact on chemistry.

Lines 267-272: I think it's important to note that CHON compounds can also be derived from $RO_2 + NO$ reactions during OH-initiated oxidation.

An acknowledgement of this pathway has now been added in lines 292-4:

'While it is also possible for these CHON compounds to originate from $RO_2 + NO$ reactions during OH-initiated oxidation, the difference in the diurnal cycle here from that observed in other locations indicates a contribution from daytime $NO_3$.'

Line 288: Can you provide some quantitative justification (i.e., data) for saying that the aerosol oxidation state in Delhi is "very low"?

We have now removed this sentence from the manuscript, and have instead included more quantitative information later in the paragraph (see our following response).

Line 293: Can you be more quantitative here (rather than just "unusually low")?

We have submitted a further paper to ACP (Huang et al., 2023, which has now been accepted for pre-print in EGUSphere) that compares the O:C ratios for aerosol from different campaigns worldwide using the FIGAERO-CIMS, including data from this campaign. Among the datasets considered, the O:C ratios from Delhi were the lowest: 0.7 vs between 0.75 and 0.95 for the other datasets (note that the iodide-CIMS is sensitive to oxygen-containing compounds, so this is likely to be much higher than O:C ratios found using other methods). We have now added some more quantitative detail in the text.

Lines 320-23: 'The average oxygen to carbon (O:C) ratio of particulate matter during this campaign has been found to be unusually low compared with a global dataset of FIGAERO-CIMS observations in different environments (Huang et al., 2023); the O:C ratio of particulate matter observed by the FIGAERO-CIMS during this campaign was 0.7, compared with values between 0.75 and 0.95 observed elsewhere.'

Line 300: Can you include a few other important details about the Beijing measurement site (was it also ground-based, was it also at a background urban site)?

The Beijing field site is closely comparable to the Delhi field site, in terms of its surroundings and sampling height. More information has now been added to the manuscript in lines 330-34:

'The sampling site for this campaign was in the west campus of the Beijing University of Chemical Technology (BUCT). Measurements were taken from the top floor of a five-storey building, at a height of approximately 20 m. The location is comparable to that of the site in Delhi: it is influenced by local pollution sources including traffic, residential heating and cooking emissions (Cai et al., 2022), and it and can similarly be considered an urban background site.'

Line 303: I think a comparison of NO emissions inventories between Delhi and Beijing is an important part of this comparison to help distinguish whether observed differences are due more to dynamics versus due more to emissions.

Although NO emissions are lower during the night in both locations, the nocturnal reduction is much smaller in Delhi, largely due to the prevalence of heavy-duty vehicles during the night, as is discussed in more detail above.

The median wintertime nocturnal boundary layer height in Beijing has been reported to be around 100 m, with some variation between 50 m and 300 m (Yang et al., 2020). In Delhi, a previous winter campaign reported a similar nocturnal boundary layer height of 100 m (Raj et al., 2021). During this campaign, however, data from READY (described above) suggest a lower nocturnal boundary layer height of 50 m. It is possible that the true value could be even lower due to difficulty measuring at this low height. This would have an impact on the relative concentration of NO between the two cities.

We therefore propose that the difference in nocturnal NO concentrations and the NO diel patterns between the two cities is a combined consequence of both emissions and dynamics. A more detailed exploration of this has now been included in the text:

Lines 350-58: 'There are two key reasons for the substantial difference in the nocturnal NO concentrations between the two cities: first, emissions inventories indicate that vehicular emissions of $NO_X$ decrease by around 90% during the night in Beijing (Jing et al., 2016), compared with a decrease of only around 40% in Delhi (Biswal et al., 2023). Due to traffic regulations restricting the movement of heavy-duty vehicles during the day, there is increased movement of these vehicles at night in Delhi (Tiwari et al., 2015), leading to greater nocturnal $NO_X$ emissions than can be found in other, comparable cities. Second, the median boundary layer height during the night in winter in Beijing has been reported to be around 100 m (Yang et al., 2020), while data from READY during this campaign (Fig. 2) indicate a nocturnal boundary layer height of 50 m or even lower. As a result, comparable $NO_X$ emissions would result in double the concentration, or even more, in Delhi compared with Beijing. Together, these factors result in the high nocturnal NO concentrations in Delhi  compared with Beijing.'

Line 305: Replace "where" with "when."

This has been updated in the manuscript.

Line 309: Because $N_2O_5$ is also thermally stabilized, a comparison of temperatures between Delhi and Beijing is another important factor to consider here. Line 325-326: I suggest adding the following italicized phrase for clarity: "*At a given pCl concentration,* larger nighttime concentrations of $N_2O_5$ result…"´

We agree that the temperature is an important factor that ought to be acknowledged, as it will have a large impact on the $N_2O_5$ concentration. We have therefore now included reference to this in the manuscript. We have also included the suggested clarification, now in lines 374-5.

Lines 346-8: 'It is important to note that the lower temperatures in Beijing (where the mean campaign temperature was around 3.4 °C, compared with 16.8 °C in Delhi), will contribute towards the higher $N_2O_5$ concentrations.'

Figure 6: Why are constrains on $O_3$ and initial $ClNO_2$ different between scenario 1 and scenarios 2 and 3?

The production of $O_3$ in Scenario 1 was used to calibrate model parameters: we adjusted the 'jcorr' value in the F0AM model, which modulates the strength of the incoming solar radiation, until the daytime peak and shape for $O_3$ matched our actual $O_3$ measurements. Comparing our $O_3$ observations with these model results suggested that there is likely a slight baseline error in our $O_3$ measurements, and that the actual average night-time $O_3$ concentration is likely very close to

zero. There appears to be a detection limit near around 2-3 ppbv. We therefore opted to use the modelled $O_3$ in this scenario, as it reduced the potential influence of such measurement artefacts. In Scenarios 2 and 3, measurements of $O_3$ were mostly above this detection limit and we therefore used the measured values in order to more accurately represent the respective scenarios.

We used the measured values of $ClNO_2$ in Scenario 1, as this represents the average for the campaign. Scenario 3 is a hypothetical scenario and we therefore started with a single, calculated value for the average night-time $ClNO_2$, and allowed this to deplete within the model. For Scenario 2, when we used measured values of $ClNO_2$ in the model, there was a slight mismatch between the time when the $ClNO_2$ began to deplete at sunrise, which resulted in $ClNO_2$ concentrations disappearing before sunrise and the model therefore being unable to form Cl radicals. One option was to shift sunrise artificially in the model for this scenario, but instead, we opted to take the same approach as for Scenario 3 and use a calculated night-time value, then allow this to deplete within the model.

Line 387: Are these large concentrations of NO a result of emissions or dynamics? This has important implications for the conclusions you later draw about emissions reductions, etc

This is the result of a combination of both of these factors. We have now updated this text in the manuscript to reflect this.

Lines 436-8: 'This is due to the presence of large concentrations of NO during the night; the result of the night-time compression of the boundary layer, coupled with large night-time emissions of $NO_x$. This nocturnal NO depletes $NO_3$ during the night and therefore results in both $NO_3$ and $N_2O_5$ consistently peaking during the day.'

References
Akimoto et al., *Atmos. Chem. Phys.*, 19, 603-615, https://doi.org/10.5194/acp-19-603-2019, 2019.

Biswal et al., *Earth Syst. Sci. Data*, 15, 2, 661-680, https://doi.org/10.5194/essd-15-661-2023, 2023.

Gulia et al., *Atmos. Pollut. Res.*, 6, 2, 286-304, https://doi.org/10.5094/APR.2015.033, 2015.

Jing et al., *Atmos. Chem. Phys.*, 16, 3161-3170, http://www.atmos-chem-phys.net/16/3161/2016/, 2016.

Nelson et al., *Environ. Sci. Technol. Lett.*, https://doi.org/10.1021/acs.estlett.3c00171, 2023.

Raj et al., *J. Geophys. Res. Atmos.*, 126, 24, https://doi.org/10.1029/2021JD035681, 2021.

Tiwari et al., *Atmos. Res.*, 157, 119-126, https://doi.org/10.1016/j.atmosres.2015.01.008, 2015.

Tripathi et al., *J. Geophys. Res. Atmos.*, 127, 12, https://doi.org/10.1029/2021JD035342, 2022.

Tyagi et al., *Atmos. Pollut. Res.*, 7, 5, 808-816, https://doi.org/10.1016/j.apr.2016.04.008, 2016.

Yang et al., *Remote Sens.* 12, 23, 3935, https://doi.org/10.3390/rs12233935, 2022.